# Effectiveness and Tolerability of a New Formulation of a Topical Anesthetic in Reduction of Pain and Parents’ Satisfaction in Pediatric Dentistry

**DOI:** 10.3390/children10030444

**Published:** 2023-02-24

**Authors:** Gianmaria Fabrizio Ferrazzano, Giuseppe Di Fabio, Roberto Gatto, Sara Caruso, Gianluca Botticelli, Silvia Caruso

**Affiliations:** 1UNESCO Chair in Health Education and Sustainable Development, Paediatric Dentistry Section, University of Naples “Federico II”, 80138 Naples, Italy; 2Department of Life, Health and Environmental Sciences, Paediatric Dentistry, University of L’Aquila, 67100 L’Aquila, Italy

**Keywords:** pediatric dentistry, child, topical anesthetic, lidocaine, prilocaine, dental injection, pain management

## Abstract

The aim of the present study was to test a new topical anesthetic gel with a different formulation (10% lidocaine, 10% prilocaine) to analyze its effectiveness in pain control, during the subsequent injection of local anesthetic, and the presence of any side effects. Methods: The study’s research design was a randomized controlled clinical trial on 300 children, aged 5–8 years, divided into two groups, each of 150 patients, according to pre-injection procedures (presence or absence of topical anesthesia). The injection pain was analyzed using the Wong-Baker Faces Pain Rating Scale (PRS) and the Face, Legs, Activity, Cry, Consolability Scale (FLACC). At the end of the procedures, patients’ parents’ satisfaction was recorded. The data were analyzed using the Student’s *T* test, Mann–Whitney U test and Chi-square test. Results: There were statistically significant differences between the two groups both in the PRS and FLACC ratings. Both in subjective and objective pain evaluations, significantly higher pain ratings were observed in the group without topical anesthesia. There was also a statistically significant difference in terms of patients’ parents’ judgment, as in the group with the use of topical anesthetic the level of parental satisfaction is statistically higher. Conclusion: The experimental anesthetic has proved very effective in its use as a topical gel in both pain measurement scales, thus validating its use on the oral mucosa, for its pharmacological and psychological effect, in the total absence of local and systemic side effects.

## 1. Introduction

The term anesthesia generically indicates the abolition of sensitivity, consciousness and pain, associated with muscle relaxation. Anesthesia is generally divided into two main strands: local and general anesthesia. Local anesthesia (LA) is a form of anesthesia used to anesthetize and thus inhibit pain on small areas of the body; it requires infiltration of an anesthetic drug through the skin or subcutaneous area to be treated. The use of local anesthesia has only specific contraindications and does not imply the loss of consciousness of the patient, who remains alert as the drugs remain confined to the affected area without exhibiting any direct action on the central nervous system.

When a correct dosage is used, the most annoying side effect associated with local anesthesia is pain at the injection site. Indeed, it is known that in an entire dental treatment the anesthetic injection is the part that causes the most fear. Local anesthesia, particularly in children, is associated with high levels of anxiety and correct administration is essential for optimal behavioral management [1]. Unfortunately, it is often necessary to use local anesthesia in children, often to treat dental caries. Indeed, dental caries is the most common chronic disease in both children and adults in the world [2]; together with orthodontic treatments, sometimes linked to important systemic pathologies [3,4,5], the therapy of carious pathology represents the most performed procedure in the pediatric population. According to Global Burden of Disease (GBD) 2017 study [1], more than 30% of children in the world have untreated caries, reaching over 50% in some states such as France and Spain. Although in the last 40 years considerable oral health prevention campaigns have been launched in Italy and numerous innovative systems are being developed to improve oral health [6], dental caries remains a significant public health problem, with a high prevalence. In 2014, in Italy, at 4 years of age, 21.6% of children have caries; at 12 years, 43.1% are affected by caries [7]. Furthermore, in other segments of the pediatric population, such as the vulnerable ones, the incidence of carious disease stands at 70.5%, including in children under 6 years of age [8].

At the time of dental anesthesia, therefore, children could typically have a negative reaction to this procedure, from the young patient’s loss of compliance to extreme reactions, in which the child becomes uncontrollable and combative, which could make it impossible for the dentist to continue with the pre-established dental treatment [9]. However, since dental fear is mainly caused both by the feeling of loss of control and by possible previous negative experiences that lead to an expectation of pain, sometimes the initial behavior of the pediatric patient is independent of our actions and, in this case, it becomes particularly important to make the anesthetic injection phase as painless as possible [10]. If we scrupulously maintain good pain control and reassure the patient, we can maximize a child’s cooperation, build a good dentist–patient relationship, and subsequent dental treatments will not be influenced by negative patient behavior, resulting in improved compliance.

To reduce the inconvenience of injection fear, especially in pediatric populations, the local anesthesia technique can be combined with topical anesthesia. Topical (or surface) anesthesia is a reversible abolition of sensitivity in a small part of the body by external and localized administration of anesthetics in the form of creams, ointments, gels or sprays, usually on an area of skin or mucosa. The intensity of anesthesia is weak, but it is easy to administer and can reduce the pain caused by needle injections, having minimal side effects, and can therefore lead to improved behavior toward dental treatment [11,12]. The effect of topical anesthetics is increased when the mucosa is dry, therefore, before the application, surfaces should be adequately dried.

The anesthetics used for the topical technique have clear differences from those used for infiltration anesthesia: the molecules used for topical anesthesia must have a high permeability of the mucosa to increase the anesthetic effect, in order to block the more superficial free nerve terminals. For this same reason, vasoconstrictors are not added to topical anesthetic mixtures because they undermine this mucosal permeability. Additionally, topical anesthetic formulations typically have higher concentrations than injectables to promote diffusion after passing through the mucosa [13]. As the concentration of anesthetics is high to facilitate infiltration and anesthesia, it should be applied in a small area to decrease the risk of toxicity. Moreover, although topical anesthetics are among the safest, dentists must always pay attention in clinical practice to any allergic reactions in patients characterized by regressed hypersensitivity reactions or neuropathies [14]. Another side effect of topical anesthetic overdose is methemoglobinemia, especially with benzocaine use; in fact, in the use of topical anesthetic sprays, Taleb stated that, in adults undergoing transesophageal echocardiography, benzocaine accounted for 66% of the total cases of methemoglobinemia compared with lidocaine (5%) and prilocaine (28%) [15]; however, methemoglobinemia associated with topical gel is rather rare, certainly in comparison with the spray. However, care should be taken not to use it in cases of age less than 3 months, prematurity, and concomitant use of a methemoglobin-inducing agent [16].

There is a wide range of applications for topical anesthesia, such as before needle insertion, as mentioned above, simple extraction of primary tooth, placement of rubber-dam clamp or orthodontic bands and to inhibit of vomiting. For this reason, topical medications can be administered in different ways, through the use of sprays, gels, solutions, ointments and patches. For each application there is a preferable way of administration [17]. Spray and solution tend to block a large area and are therefore mainly used with patients with a strong urge to vomit, before taking x-rays or taking impressions. As a negative effect, being distributed over a fairly large area, they can increase the risk of absorption in the circulatory system [15]. Gels and ointments are mainly used in case of lacerations and abrasions or before needle insertion. Lidocaine is the most common and used topical anesthetic and has prompt and fast action. Among the topical anesthetics, Prilocaine is the one considered to be the best tolerated and, as it has no vasodilating action, unlike Lidocaine, it has a greater persistence in the tissues. Benzocaine is characterized by slow absorption and, to have a sufficient anesthetic effect, it is used in higher concentrations, ranging from 10 to 20% [18,19,20,21]. Alternatively, mixtures of topical anesthetic agents are used as topical anesthetics and they are called eutectic mixtures of local anesthesia (EMLA) [22]. For example, in dermatology, EMLA cream (a 1:1 mixture with 2.5% prilocaine and 2.5% lidocaine), has been widely used as a topical skin anesthetic since the 1980s [23].

The aim of this study was to test a new topical anesthetic gel with a different formulation (10% lidocaine, 10% prilocaine) to analyze the effectiveness in pain control and the presence of any side effects.

## 2. Materials and Methods

### 2.1. Data Source

The study’s research design was a randomized controlled clinical trial with children, aged 5–8 years, conducted from February 2020 to December 2021. A minimum sample size of 267 subjects was determined via the G*Power software program (power = 0.80, α = 0.05, β = 0.20, G*Power Ver: 3.1.9.2). At the beginning of the study, a total of 333 children were recruited, calculating possible losses due to the exclusion criteria. The protocol of the study was approved by the Research and Ethics Committee of the University of Naples Federico II (Prot. N. 23/2019).

The operations were conducted in the Pediatric Dentistry Center in Sedation of Naples and supported by the UNESCO Chair in Health Education and Sustainable Development of ‘Federico II’ University of Naples between February and September 2022 by six pediatric dentist operators. All examiners were calibrated on pain recording at the UNESCO Chair of Naples and the kappa test revealed a final score of k = 0.90 (CI 95% 0.795–0.952).

All patients had never undergone infiltrative anesthesia. We selected patients who needed treatment that required anesthesia, but without pain at the time of the first visit. Therefore, we excluded patients who presented inflammatory or infectious processes (such as irreversible pulpits or abscesses) at the time of the treatment. Among the exclusion criteria based on medical history, there was also a history of allergies, the presence of chronic pathologies and the use of medications (such as NSAIDs or corticosteroids) at the time of treatment. After exclusion, the final sample size was 300 children.

A written informed consent, with an explanation of potential risks and benefits, was discussed and obtained from each parent of all eligible children who agreed to participate.

### 2.2. Individual Variables

In this randomized clinical research, patients were divided into two groups, each of 150 patients, according to pre-anesthesia procedures (children were randomly assigned to two groups by lottery method, assigned to each patient with a computer-assisted program):Group A: (topical anesthesia group) The injection site has dried out, and experimental strawberry-flavored topical anesthetic gel (containing 10% lidocaine, 10% prilocaine and mucoadhesive gel q.b.) was applied with a cotton pellet, using moderate pressure with rubbing circular motion for 30–45 s to a confined site and left for about 5 min, before local injection.Group B: (no topical anesthesia group).

Each patient received an injection of LA solution (Ultracain D-S forte; Hoechst Canada 145 Inc., Montreal, Canada) using a 27-gauge dental needle, as follows: the patient opened the mouth, and, using reassuring language, the patient’s lip was lifted keeping the tissue taut, the needle was inserted parallel to the long axis of the tooth at the level of the mandibular or maxillary fornix, the injection was carried out at a flow rate of 1 mL/min and, finally, the needle was gently removed.

The effectiveness of pre-anesthesia methods on injection pain was assessed subjectively and objectively and then a judgment was asked of the children’s parents.

Subjective assessment: The WBFRRS, or Wong-Baker Faces PRS (Pain Rating scale) was used for subjective assessment [24]. It is a one-dimensional pain rating scale used for children between 3 and 8 years of age. It is based on the indication by the child of a face, among a series of 6 drawn faces, which reflects the intensity of the pain he is experiencing at that moment. Values range between 0 and 10: 0 signifies “no hurt” and 10 indicates “hurts worst”.

Objective assessment: The FLACC (Face, Legs, Activity, Cry, Consolability Scale) scale was used for objective assessment [25]. The FLACC scale is a pain measurement scale based on the observation of the child’s behavior. Each of the 5 parameters included in the scale (face, legs, activity, crying and consolability) has three descriptors, which can be assigned a score between 0 and 2, generating a total score between 0 and 10. Based on the score obtained, the child’s pain can be quantified: 0 means “comfortable, no pain”, 1–3 means “mild pain”, 4–6 means “moderate pain”, 7–10 means “severe suffering or pain”.

Patients’ parents’ judgment: at the end of the procedures, parents, who remained in the same room as the child for the duration of the treatment session, were asked whether or not they were satisfied with the procedure and their answers were recorded.

### 2.3. Statistical Analysis

Data obtained were statistically analyzed with Stata software (Stata Corp LP, College Station, TX, USA). The Student’s *T* test was used to evaluate the differences between groups, in terms of both pain scales ratings. The relationship between pain ratings and gender was evaluated with the Mann–Whitney U Test. The Chi-square test was used to compare the distribution of parents’ patient judgment. The significance level (α) was set to 0.05. Only patients with complete data on all analyzed variables were included in the analysis; those with incomplete data were excluded.

## 3. Results

The trial was completed with good compliance. No allergic reactions or side effects were observed.

Three hundred children—142 girls (47%) and 158 boys (53%)—aged between 5 and 8 years (6.37 ± 0.46) were included in this study. The relationship between pain ratings and gender has not been included in the tables, as the Mann Whitney U test showed that there are no statistically significant differences (*p* > 0.05) in both the PRS and FLACC scales.

In both pain rating scales, PRS (*p* < 0.001) and FLACC (*p* < 0.001), showed statistically significant differences between the two groups (Table 1 and Table 2). In particular, in both tables, Group B (No topical anesthesia) showed higher pain ratings.

Distribution of the PRS scale pain ratings is shown in Figure 1 and FLACC scale pain ratings in Figure 2.

On the Wong-Baker PRS scale, the mean score of Group A (1.83 ± 2.31) was lower than that of Group B (4.35 ± 2.60). More precisely, the number of “no hurt” ratings was lower in Group A than in Group B (A: 65; B: 30) and the number of “hurts worse” ratings was lower in Group A than in Group B (A: 65; B: 30) (Figure 1).

Likewise, on the FLACC scale, the number of “no pain” ratings was higher in Group A than in Group B (A: 65; B: 30). In addition, the number of “severe pain” ratings was lower in the Group A than in Group B (A: 20; B: 50) (Figure 2). Thus, lower mean values were found in Group A (2.22 ± 0.23) than in Group B (4.50 ± 3.08).

There was also a statistically significant difference in terms of parents’ patient judgment, according to the Chi-square test (*p* < 0.001). As shown in Table 3, in Group A, the majority (83%) of parents are satisfied with the procedure; in Group B the percentage of satisfied with the procedure decreases, attesting to 40%.

## 4. Discussion

The purpose of the present study was to test a new formulation of topical anesthetic to analyze its effectiveness in pain control, during the subsequent injection of local anesthetic, for decreasing the discomfort that characterizes injection with a local anesthetic on a fairly large sample of patients. In fact, often, especially in the pediatric population, the patient comes to the dental visit for treatment rather than for preventive reasons, already having a pathology in progress, especially the carious one; hence, there is no opportunity to create good compliance before having to perform anesthesia. Therefore, although numerous innovative discoveries have been made in dentistry for the treatment of carious pathology and its consequences [26,27], in the field of local anesthesia the methods are more or less always the same. The injection is the procedure that still causes the most fear and anxiety in a dental session.

The reported effects of an alternative mixture of topical anesthetics (such as EMLA) in the literature have been inconsistent and have divergent results, Many providers are using those for periodontal and orthodontic procedures instead of traditional local anesthetic injections [17,28,29,30]. In a study by Milani et al. [31] and a study by Abu et al. [32], EMLA (2.5% lidocaine and 2.5% prilocaine) was found to be significantly more effective in reducing pain of maxillary infiltration injections than 20% benzocaine. Reznik et al. [30] found that a CTA containing lidocaine 20%, tetracaine 4%, and phenylephrine 2% was more effective than 20% benzocaine in pain reduction during placement of orthodontic temporary anchorage devices. In a study by Primosch and Rolland [33], EMLA and 20% benzocaine were found to be equal in pain reduction for palatal injections in children. Meanwhile, Tulga and Mutlu [34] observed 20% benzocaine to be more effective than EMLA for pain reduction of dental injections in their study of pediatric subjects.

In this study, the results presented in Table 1, Table 2 and Table 3 show an excellent level of pain control compared to the control group. The level of satisfaction of the parents who attended the session together with the pediatric patient was also of high level. The observations presented in this study, therefore, underline that, both in the PRS and in the FLACC, the percentage of “no hurts” subjects is 43% in the test group, compared to 8% and 20%, respectively, in the control group. This data is of fundamental importance especially in the pediatric population, as in this category of dental patients, the presence of discomfort, albeit minimal, is managed differently than in the adult population, creating children who will view future dental visits negatively. The use of this particular formulation of topical anesthetic meant that, in this sample of patients, almost half of the test group did not have the slightest perception of the injection of the local anesthetic, with the creation of immediate compliance between the operator and the patient, being able to complete the treatment without any problem. Furthermore, even in the highest pain classes in the PRS (5–6/7–8/9–10) and FLACC (4–6/7–10) scales, there are statistically significant different results. In fact, in the PRS scale, the differences are very marked, since in the test group the percentage of patients who respond discomfort from “even more” to “hurts worst” is 14% (5–6: 8%; 7–8: 5%; 9–10: 1%) while in the control group the data are not positive as 48% of patients fit into the highest pain values (5–6: 30%; 7–8: 9%; 9–10: 9%). For the FLACC scale, also, only 30% (4–6: 17%; 7–10: 13%) are in the higher classes as regards the group treated with topical anesthetic, while in the control group the percentage is much higher, reaching at 63% (4–6: 30%; 7–10: 33%).

All these data are then confirmed in the satisfaction questionnaire delivered to the parents of the patients: the percentage of parents not satisfied with the procedure in the group with the experimental topical gel is only 17%, while in the group with exclusively infiltrative anesthesia the unsatisfied parents comprise as much as 60%. All these results occurred with the total absence of both local and systemic side effects in all patients to whom the experimental formulation was administered and, moreover, operators noted that the strawberry flavor made it particularly pleasant to use for children in group A.

Pain management during the injection of local anesthetic is a fundamental step of treatment in a pediatric patient and, when it cannot be avoided, always represents a critical moment for dentists, as the outcome of the anesthesia will influence the relationship with the young patient, who, in the event of a positive outcome, will have greater confidence in subsequent therapeutic sessions.

In this study we evaluated the use of a new formulation of topical anesthetic, using the main pain assessment scales as parameters. The data obtained proved to be satisfactory with excellent values in terms of the efficacy of the anesthetic in reducing pain, compared with the control group, and the total absence of side effects. Further studies will be needed, especially by comparing this formulation with others on the market (such as 20% benzocaine or, particularly, EMLA cream) and with a placebo group, in order to validate its possible superiority over others and to show a dose-response effect, i.e., whether the 10%/10% gel worked better than the 2.5%/2.5% gel.

## 5. Conclusions

The experimental anesthetic has proved very effective in its use as a topical gel both in the subjective scales and in the objective pain measurement scales, thus validating its use on the oral mucosa, for its pharmacological and psychological effect, in the total absence of both local and systemic side effects. Therefore, the use of a topical anesthetic should always be recommended before performing a local anesthetic injection in the pediatric population, to decrease discomfort and for optimal behavioral outcomes. Furthermore, the procedure led to a high level of parental satisfaction.

## Figures and Tables

**Figure 1 children-10-00444-f001:**
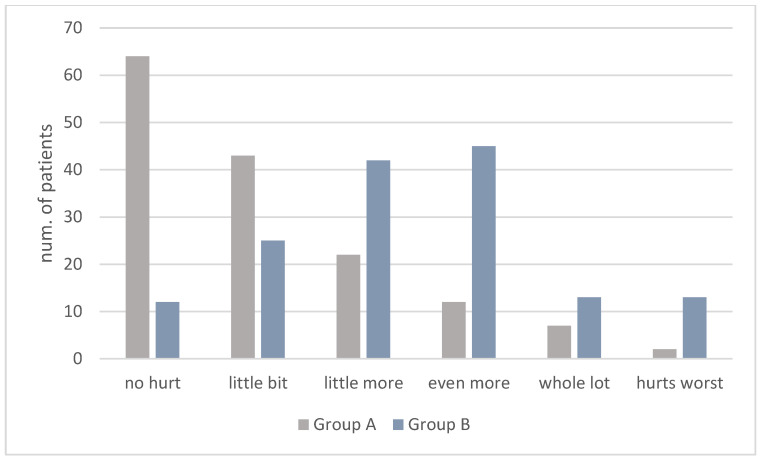
PRS scores in Group A and B.

**Figure 2 children-10-00444-f002:**
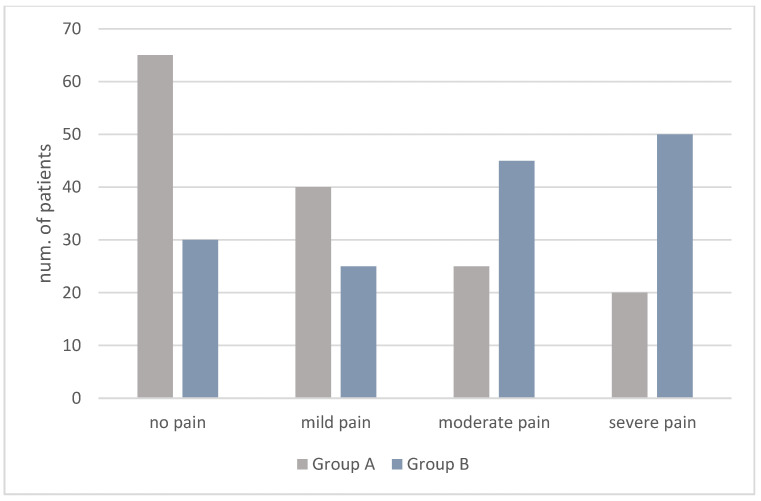
FLACC scores in Group A and B.

**Table 1 children-10-00444-t001:** Distribution of the PRS scale pain ratings.

PRS Scores								
	n.	0	1–2	3–4	5–6	7–8	9–10	Mean ± SD
Group A	150	64 (43%)	43 (29%)	22 (15%)	12 (8%)	7 (5%)	2 (1%)	1.83 ± 2.31
Group B	150	12 (8%)	25 (16%)	42 (28%)	45 (30%)	13 (9%)	13 (9%)	4.35 ± 2.60

*p* < 0.001 statistically significant with Student’s *T* test.

**Table 2 children-10-00444-t002:** Distribution of the FLACC scale pain ratings.

FLACC Scores						
	n.	0	1–3	4–6	7–10	Mean ± SD
Group A	150	65 (43%)	40 (27%)	25 (17%)	20 (13%)	2.22 ± 0.23
Group B	150	30 (20%)	25 (17%)	45 (30%)	50 (33%)	4.50 ± 3.08

*p* < 0.001 statistically significant with Student’s *T* test.

**Table 3 children-10-00444-t003:** Distribution of parents’ patient judgment.

Parents’ Patient Judgment			
	n.	Satisfied	Not-Satisfied
Group A	150	125 (83%)	25 (17%)
Group B	150	60 (40%)	90 (60%)

*p* < 0.001 statistically significant with Chi-square test.

## Data Availability

Data will be made available upon reasonable request to the authors.

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
