# Peer review of "Effectiveness and Tolerability of a New Formulation of a Topical Anesthetic in Reduction of Pain and Parents’ Satisfaction in Pediatric Dentistry"

_children, 2023, doi:10.3390/children10030444_

Round 1

Author Response

Good morning. I have attached the corrections point-by-point. Thank you

Reviewer 2 Report

Interesting topical formulation.  There are a couple of references that should be added concerning the lidocaine patch (which clearly works well in needle insertion pain and also demonstrates that higher concentrations must be used (in this case 10% and 20% lidocaine than for infiltration injections 2%.  Also you forget about the use of topicals for toothache and I have given you a reference or two (there are others)

In a number of places I changed the way you worded things for better comprehension among the readership.  Instead of me just repeating what I annotated in your manuscript I will send you the manuscript with my annotated comments and edits.  I also bring up the possibility of methemoglobinemia with excessive doses of prilocaine.  It has happened with injectable formulations.  

Author Response

(The authors gave the same response as above.)

Round 2

Reviewer 1 Report

Most of my suggestions were applied and I have nothing more to add concerning the writing which is good.

However, my main methodological concern was not adressed, and even worsened.

You just deleted the part about beta risk without adressing it. Beta risk or power should be mentionned, be correct, and be used correctly when running statistical analysis.

You changed the methodology to mention the suggested test and added the p values but those p values are incoherent with what you observed. I am unable to replicate you analysis using student's test because the full data are not available; but I could replicate the chi2 test on two different softwares and the result you should get is around 1.17e-14; much lower than the reported 0.008. Student's test should give results around the same magnitude. 

I also noted that in table 1; your B group sums for 160 patients, not 150.

Finally, table 1 and 2 are reporting continuous values as discrete (which is bad practice but not unacceptable) and I suspect you mixed continous and discreet data as well as qualitative and quantitative statistical tests. 

Overall, there are so many inconsistencies and flaws in the statistical analysis that the results can't be trusted.

I advise you to seek out a statistician to help you for the analysis.

Author Response

I am attaching the changes made regarding materials and methods and the results (together with the tables made by the statistician).
As for the beta risk and power part, it should be written in Data source, but if you want I will also repeat it in the statistical analysis part.
As for statistical significance, I had the statistician retest using STATA software and the p-values come out much lower (as you indicated).
I also corrected the typo in the table and changed some other values slightly..
Thanks a lot for the corrections.
